# Laser Cleaning Improves Stem Cell Adhesion on the Dental Implant Surface during Peri-Implantitis Treatment

**DOI:** 10.3390/dj11020030

**Published:** 2023-01-20

**Authors:** Taras V. Furtsev, Anastasia A. Koshmanova, Galina M. Zeer, Elena D. Nikolaeva, Ivan N. Lapin, Tatiana N. Zamay, Anna S. Kichkailo

**Affiliations:** 1Department of Orthopedic Dentistry, Faculty of Dentistry, Krasnoyarsk State Medical University, 1 Partizana Zheleznyaka, Krasnoyarsk 660022, Russia; 2Laboratory for Biomolecular and Medical Technologies, Krasnoyarsk State Medical University, 1 Partizana Zheleznyaka, Krasnoyarsk 660022, Russia; 3Federal Research Center “Krasnoyarsk Science Center” of the Siberian Branch of the Russian Academy of Sciences, 50 Akademgorodok, Krasnoyarsk 660036, Russia; 4Department of Materials Science and Materials Processing Technology, Polytechnical Institute, Siberian Federal University, Pr. Svobodniy, 79, Krasnoyarsk 660041, Russia; 5Laboratory of Advanced Materials and Technology, Siberian Physical Technical Institute, National Research Tomsk State University, Tomsk 634050, Russia

**Keywords:** peri-implantitis, dental implants, laser, titanium brush, mesenchymal stem cells, adhesion

## Abstract

Dental implant therapy is a well-accepted treatment modality. Despite good predictability and success in the early stages, the risk of postplacement inflammation in the long-term periods remains an urgent problem. Surgical access and decontamination with chemical and mechanical methods are more effective than antibiotic therapy. The search for the optimal and predictable way for peri-implantitis treatment remains relevant. Here, we evaluated four cleaning methods for their ability to preserve the implant’s surface for adequate mesenchymal stem cell adhesion and differentiation. Implants isolated after peri-implantitis were subjected to cleaning with diamond bur; Ti-Ni alloy brush, air-flow, or Er,Cr:YSGG laser and cocultured with mice MSC for five weeks. Dental bur and titanium brushes destroyed the implants’ surfaces and prevented MSC attachment. Air-flow and laser minimally affected the dental implant surface microroughness, which was initially designed for good cell adhesion and bone remodeling and to provide full microbial decontamination. Anodized with titanium dioxide and sandblasted with aluminum oxide, acid-etched implants appeared to be better for laser treatment. In implants sandblasted with aluminum oxide, an acid-etched surface better preserves its topology when treated with the air-flow. These cleaning methods minimally affect the implant’s surface, so it maintains the capability to absorb osteogenic cells for further division and differentiation.

## 1. Introduction

To date, dental implant therapy replacing missing teeth is a well-accepted treatment modality. Despite good predictability and success, especially in the early stages after the surgery, the risk of inflammation and post-replacement complications occurring both in the early and long-term periods remains an urgent problem [1]. Peri-implantitis is an inflammatory process affecting hard and soft tissue around an implant, leading to the loss of supporting bone [2,3].

Peri-implantitis is one of the leading and frequent reasons for dental implant rejection, which goes through four stages in its development. The first stage is characterized by a slight loss of bone tissue in the horizontal direction. During the second stage, there is a moderate bone height decrease with the vertical defect in the area of the connection between the implant and the bone happens. At the third stage of peri-implantitis, there is a moderate bone height reduction with a vertical defect along the entire implant occurs. The last stage is characterized by bone resorption of the alveolar process.

The main etiological factor in the development of peri-implantitis diseases is a biofilm, which is a complex microbial community of numerous microorganisms [4,5]. Biofilm contains more than 700 different types of microorganisms, develops on the surface of the implants, and leads to peri-implant mucositis and finally to peri-implantitis [6]. It has been shown that bacterial colonization begins as early as 30 min after implant placement, and after two weeks, this biofilm is already well organized in the peri-implant space [6]. In subsequent months, it changes qualitatively [7]. Gram-positive cocci and immobile bacilli predominate on healthy implants. Mucositis, cocci, motile bacilli, and spirochetes predominate in peri-implant. Peri-implantitis is characterized by Gram-negative, motile, and anaerobic species. Pathogenic microbiota causes inflammation and implant rejection [5,6].

As nonsurgical peri-implantitis antibiotic therapy is minimally effective [8], surgical access and decontaminating the infected implant surfaces with various methods are usually required. Chemical and mechanical methods include titanium brushes, air-power abrasives, ultrasonic use, lasers, dental burs, saline, chlorhexidine, and hydrogen peroxide wash [9,10]. Many scientific studies are devoted to treating peri-implantitis [11,12,13,14,15]. Ideally, the cleaning method should minimally affect the micro-roughness of the dental implant surface, initially designed for good cell adhesion and bone remodeling [8], and provide full microbial decontamination [16,17,18].

In this regard, clinical studies of the treatment of peri-implantitis, where the treatment of the implant surface is performed with lasers, are of great interest and perspective. Several studies indicate a more predictable result of therapy when a laser is used to treat the infected surface of the implant and then regenerative techniques for bone restoration. This makes it possible to eliminate the marginal gums’ inflammation and the bone pocket’s complete disappearance. Most studies are about Er:YAG, CO_2_, and diode laser [19,20].

The most important long-term outcome is bone regeneration after peri-implantitis treatment [15,21]. Therefore, the surface should promote cell regeneration [22,23]. Data on the in vitro study of cell adhesion ability on various dental implant surfaces subjected to different processing methods during peri-implantitis are minimal. After the surgical access and cleaning of the implant surface from the bacterial film, it surrounds blood cells and bone marrow cells, forming a temporary bioactive layer. In the second phase, osteoclasts trigger bone tissue absorption a month later. Mesenchymal stem cells (MSC) and osteoblasts migrate to the osteogenic area on the implant’s surface and start proliferating, differentiating, and initiating the mineralization procedure. Approximately three months after the implantation (in phase three), osseointegration is in progress, and the implant surface is surrounded by osteoblasts and osteocytes (Figure 1). Mesenchymal stem cells (MSC) stimulate new bone formation during peri-implantitis treatment [24].

Peri-implantitis is a formidable complication that causes problems with rehabilitating patients with adentia. The solution to this problem will increase the effectiveness of implant prosthetics and achieve even greater patient loyalty to adopting a treatment plan on implants. It also allows dentists to make more predictable decisions based on objective scientific findings. In connection with the preceding, the search for optimal and predictable methods for treating peri-implantitis remains relevant.

A complete peri-implantitis treatment is stepwise: the dental plaque has to be removed, and inflammation has to be eliminated. A complex therapy includes maximally effective cleaning of the implants’ surface. The current study investigates the influence of four different cleaning methods on the effectiveness of osseointegration in three different commercial implants.

## 2. Materials and Methods

The Local Ethics Committee approved MSC isolation and animal experiments at the Krasnoyarsk State Medical University (#77 from 26 June 2017). The Local Ethics Committee at the Krasnoyarsk State Medical University (#4 from 12 October 2021) approved experiments with the patients’ isolated implants.

### 2.1. Implants Preparation

Implants isolated from the patients with peri-implantitis after 3–10 years of functioning were used to examine the possibility of MSC to attach to implants surface after different cleaning methods.

Implants from three manufacturers with different surfaces were studied: 1—NobelBiocare (Gothenburg, Sweden), TiU-nite surface (anodized titanium dioxide); 2—XIVE Sirona Dentsply (Praha, Czech Republic), SLA surface (sandblasting with aluminum oxide, acid etching); 3—BioHorizons (Birmingham, AL, USA), RBM surface (tricalcium phosphate blasting/acid etching).

New implants from the manufacturer’s packaging were taken as control.

Implants isolated after peri-implantitis were subjected to the following types of cleaning: 1—diamond bur (grain size–red marking) prepared the surface of the implant under water–air cooling until the turns were thoroughly abraded; 2—with a special brush for cleaning the surfaces of implants made of Ti-Ni alloy (Neobiotec), the entire surface was treated under water–air cooling until the top layer was completely removed so that the surface became smooth but with the preservation of coils; 3—Er,Cr:YSGG laser with a wavelength of 2780 nm with the following characteristics: power 1.5 W, frequency 20 Hz, water/air 80/80, tips SF18. Uniform movements over the entire surface of the implant were processed under visual control until the complete disappearance of contaminants. 4—Air-Flow Handy 3.0 perio (EMS), Erythritol filler, with the following characteristics: air pressure 2.2 bar; cleaning water flow 41.5 mL/min, continuous treatment of the entire surface for 2 min, visual control until complete cleaning the whole surface.

All experiments were performed in triplicates: three implants from each studied manufacturer isolated from the patients with peri-implantitis were cleaned using every method. Three new implants from each manufacturer were used as a control.

### 2.2. Evaluation of the Ability of MSCs to Attach to the Implant Surface In Vitro

In the next step, we estimated the ability of MSC to attach to the implant’s surface in vitro. Implants isolated after peri-implantitis were cleaned by diamond dental burr (n = 3), titanium brush (n = 3 for each manufacture), air-flow (n = 3 for each manufacture), or laser (n = 3 for each manufacture); untreated implants (n = 3 for each manufacture) were taken as a positive control. Implants, wormed in 37 °C phosphate buffer, were placed in a 3 cm petri dish, and then cells in a GrowDex polymer matrix were planted on the top of the implants and wholly covered with the medium. MSC was cocultured with the implants for five weeks (Figure 2). The experimental procedure was repeated three times.

### 2.3. Scanning Electron Microscopy

The study of the morphology and elemental composition of the implant surfaces depending on the cleaning method compared to the control implants was performed by electron microscopic methods. Electron microscopy studies were carried out in the laboratory of electron microscopy of the Center for Collective Use of the Siberian Federal University on a scanning electron microscope (SEM) JEOL JSM 7001F (JEOL Ltd., Akishima, Japan). The surface morphology was studied in the secondary electron mode (sei) at ×1500 and ×5000 magnifications.

The JEOL JSM 7001F is equipped with an INCAPentaFETx3 energy-dispersive spectrometer (Oxford Instruments, Oxford, UK), which allows for analyzing chemical elements from B (boron) to U (uranium). The X-ray spectral microanalysis (EDX) method with a high degree of locality makes it possible to determine the elemental composition of an object. Before studying the elemental composition of the samples, the spectrometer was calibrated against Co at an operating accelerating voltage of 15 kV. X-ray spectral microanalysis was carried out at an accelerating voltage of 15 kV, a beam current of 7 × 10^−9^ A, and a working distance of 10 mm.

### 2.4. Obtaining Primary Cultures of Mesenchymal Cells from Mouse Bone Marrow

Mice weighing 25–30 g were subjected to a cervical spine dislocation. The femur and tibia were removed, freed from soft tissues, and washed with alcohol and saline. The epiphyses of the obtained bones were separated. The diaphysis of the femur and tibia were washed with phosphate buffer containing 100 units/mL of penicillin and 100 μg/mL of streptomycin. The resulting suspension was washed by centrifugation with DMEM containing 15% fetal bovine serum, 100 U/mL penicillin, 100 μg/mL streptomycin, and 12 mM L-glutamine (growth medium). To do this, the suspension was placed in conical tubes containing 5 mL of growth medium and centrifuged for 10 min at 400× *g*. The supernatant was removed. The pellet from each tube was resuspended in the growth medium to a concentration of 1 × 10^6^ cells/mL. The resulting cells were planted on 3 cm Petri culture dishes with 4 mL of suspension per dish. Cultivation is carried out in a CO_2_ incubator at 5% CO_2_ and a temperature of 37 °C. After 48 h, non-attached cells were removed, and 7 mL of fresh growth medium was added to the vials. The attached cells were cultured for two weeks, changing the growth medium every 3–4 days. Upon reaching the monolayer, the cells were washed two times with a 0.25% solution of trypsin mixed with Versen solution in a ratio of 1:1. Then, 2 mL of the above mixture was added to the vial and incubated at 37 °C for 5–10 min. Cells were transferred into suspension by pipetting.

The resulting cell suspension was precipitated by centrifugation at 400× *g* for 5 min. Implants were placed in a 3 cm petri dish, which was previously prepared and heated in phosphate buffer to 37 °C. Implants were placed with the treated side up. The pellet of the cell suspension is resuspended in a growth medium diluted with a GrowDex polymer matrix (UPM Biomedicals, Helsinki, Finland) in a ratio of 2:1 (final cell concentration 10^4^ cells/mL). The resulting cells were planted in 3 cm culture Petri dishes with implants, 3.5 mL of suspension per dish, and another 1.5–2 mL of nutrient medium with a GrowDex polymer matrix. Cultivation was carried out for 3–4 weeks until the cells sat on the implant and began to divide well. Cell growth and division were recorded using an inverted microscope.

### 2.5. Laser Scanning and Fluorescence Microscopy

Cell nuclei staining was performed using the fluorescent dye Hoescht33342 (at a final concentration of 5 μg/mL). Microscopy was performed using ZOE (Bio-Rad Laboratories, Hercules, CA, USA) fluorescent cell imaging systems and LSM 780 NLO (Carl-Zeiss, Oberkochen, Germany), ×20, 40 magnification.

### 2.6. Determination of the Presence of Cells on the Surfaces of Implants by Flow Cytometry

After microscopy of implant samples, each implant was separately removed from the culture medium and placed in a microtube with acutase solution preheated to 37 °C for 2 min (this is enough to remove cells from the surface). After that, the cells were thoroughly washed off the implants by repeated washing with pipetting. The implant was washed in phosphate buffer, the cells were removed from the surface were sedimented by centrifugation, and culture medium DMEM with 10% FBS was added to stop the action of acutase. Cells were counted using flow cytometry on a Cytomics FC 500 cytometer (Beckman Coulter, Indianapolis, IN, USA). The number of cells in the sample was calculated by analyzing their concentration in a specific sample volume taken for analysis.

### 2.7. Cell Differentiation Assay

Pre-osteoblasts, osteoblasts, and further osteocytes differentiated from MSC start to accumulate Ca_3_(PO_4_)_2_, which can be detected using an alizarin red staining kit, according to manufacturers.

### 2.8. Statistical Analyses

The significance of differences in the number of cells adherent to the implants’ surface was calculated using Welch’s t-test (a variant of the t-test for the case of different variances in groups).

We were interested in the cleaning method with the closest results on cell adhesion to the implants with the new surface. Therefore, we searched for the most “indistinguishable” pairs of groups and calculated the *p*-value for this pair of groups.

2023Figures were created with BioRender.com, accessed on 26 May 2022.

## 3. Results

The surfaces of dental implants play a crucial role in osseointegration. Therefore, maintaining the original implant surface during peri-implantitis treatment is essential to preserve their osteogenic potential [25]. Various implant purification methods differently modify the surface. In this work, we compared the effect of the implant treatment method and the efficiency of populating the implant surface with mesenchymal stem cells modified in various ways. For this, three types of structures were compared: NobelBiocare (Gothenburg, Sweden), TiU-nite shell (anodized titanium dioxide); XIVE Sirona Dentsply (Praha, Czech Republic), SLA surface (sandblasting with aluminum oxide, acid etching) and BioHorizons (Birmingham, AL, USA), RBM surface (tricalcium phosphate blasting/acid etching). Since an essential condition for osseointegration is the creation of optimal conditions for the functioning of our own mesenchymal stem cells and an increase in their population, one of the objectives of this study was to compare the efficiency of colonization of implants with different surfaces by mesenchymal cells.

### 3.1. Scanning Electron Microscopy

Different treatment methods were applied to clean the surfaces of the implants isolated from the patients because of peri-implantitis. According to scanning electron microscopy analyses, the surfaces of the new implants dramatically differ from those of the implants cleaned during peri-implantitis treatment. Different cleaning methods influence the surface topology of the implants and could cause pore cell adhesion to increase recovery time after peri-implantitis treatment. Unlike the other techniques, laser treatment helped save the overall structure and cavities on the TiU-nite and SLA surfaces. Dental Bur and titanium brush destroy the initial surfaces of all implants (Figure 3).

### 3.2. Mesenchymal Stem Cells Adhesion on the Surfaces of the Differently Treated Implants

The ability of MSC to attach to the new and cleaned implant surfaces was estimated using laser scanning microscopy and flow cytometry.

The osteogenic potential of new implants depends on their coating (1) anodized with titanium dioxide (TiUnite); (2) sandblasted with aluminum oxide, acid-etched (SLA); and (3) tricalcium phosphate blasting/acid etching (RBM). Studies have shown that titanium dioxide anodized surface is the most optimal for adhesion, division, and differentiation compared to other types of implants. The number of osteogenic cells on the implant coated with titanium dioxide was almost 20% higher than on the others. The highest number of MSC was observed on the new untreated surface of new TiUnite implants with a smaller number of cells attached to the feelings of SLA and RBM implants (Figure 4).

The number and density of the attached cells on the implants isolated after peri-implantitis depended on the cleaning method (Figure 4). MSC did not adhere well on the covers of all implants treated with a dental bur (Figure 4). Treatment with a titanium brush was similar (Figure 4). This is probably because the implant’s surface was destroyed according to electron microscopy results (Figure 2(B1,B2,C1,C2)). Laser and air-flow maximally preserved the implant surface and MSC attached to these surfaces. Cell adhesion was better for the laser-treated TiUnite and SLA implants (Figure 4). RBM implants had better cover after air-flow treatment than after the laser (Figure 2(D1,E1)). Cell culture data (Figure 4) supported electron microscopy imaging (Figure 2). The number of MSC attached to the implant’s surfaces depended on the cleaning methods (Figure 4).

MSC adhesion on different surfaces was proved by laser-scanning microscopy (Figure 5). The nucleus of the cells, attached to the surface, was stained with the fluorescent dye DAPI. It is seen that MSC prefer to adhere to micro-roughness surfaces; fewer cells are attached to smoothed ones (Figure 5). Anodized with titanium dioxide (TiUnite) and sandblasted with aluminum oxide, acid-etched (SLA) surfaces appeared to be better for the laser treatment, as MSC were more evenly distributed in caverns preserved after the cleaning. As a surface sandblasted with aluminum oxide, acid-etched (SLA) better preserves its topology when treated with air-flow.

The differentiation rate was estimated by the ability of the cells to accumulate Ca_3_(PO_4_)_2_ after five weeks of coculture of the implants with MSC. Control MSC did not stain the alizarin red staining kit and thus were not differentiated. Most cells on the new, air-flow, and laser-treated implants were differentiated and accumulated calcium phosphate to a greater extent than cells on titanium brush-treated implants. Cells in dental bur-treated surfaces stayed undifferentiated after five weeks of experiments.

## 4. Discussion

To achieve good osseointegration of the implant into surrounding tissues, it is necessary to attract osteogenic cells, such as osteoblasts and mesenchymal stem cells, to the implant’s surface. The effect of mesenchymal stem cells is due not only to their ability to maintain a regenerative microenvironment in damaged tissues but also to reduce the level of TNF-α, which blocks osteoblast differentiation. Thus, mesenchymal stem cells indirectly contribute to the activation of osteoblast differentiation and tissue regeneration. Creating the necessary concentration of stem cells and conditions for their osteogenic differentiation and transportation to the site of reparative resurrection are prerequisites for osseointegration. Cell adhesion is the first stage of interaction between cells and implants, which consists of four stages: protein adsorption, contact of cells with the material, attachment, and distribution. The quality of adhesion is critical to the ability of cells to proliferate and differentiate. Therefore, the quality of the implant surface is the most crucial factor in osseointegration.

The implant’s surface is in contact with the liquid component of the oral cavity, so its chemical, physical, mechanical, and topographic characteristics play a decisive role in successful osseointegration [25].

One of the most common materials for the fabrication of implants is titanium, which, despite its high strength, excellent biocompatibility, low toxicity, and high corrosion resistance in its pure form, is highly reactive. Interaction with oxygen stabilizes the titanium surface layer by forming titanium oxide TiO_2_. Anodizing increases the thickness of the TiO_2_ layer of the implant and gives it a moderate roughness, improving osteoconductivity and osseointegration [26,27]. Therefore, implants with anodized surfaces show the best survival rate (98.5%) with a follow-up period of at least ten years [28].

The methods used to modify the surface of titanium-based alloys make it possible to change the surface roughness and its morphological characteristics in a wide range of parameters. These possibilities make it possible to successfully use various methods of surface nanostructuring when creating dental implants based on titanium and its alloys. It has been shown that anodized surfaces have the highest number of hydroxyl groups compared to surfaces treated with sandblasting and acid etching and contact with the hydrophilic structures of saliva, tissue, and blood [28,29]. In addition, hydrophilic surfaces can support the better adhesion of mesenchymal stem cells, promoting the attachment of soft gingival tissues with a functional biological contraction to prevent microbial colonization [28,30]. Our studies confirmed that an anodized TiO2 surface is the most optimal of all those studied for stem cell adhesion.

Three types of implants were used in our work: titanium NobelBiocare with anodized TiUnite surface, XIVE Sirona Dentsply with SLA surface treated with aluminum oxide sandblasting and acid etching, and BioHorizons with RBM surface treated with tricalcium phosphate blasting and acid etching. The character of titanium implants treated with anodization has tubular nanostructures with high biocompatibility; those most suitable for cell adhesion properties can be available by changing treatment parameters. Acid etching disinfects, increases the surface area, and improves the contact of bone with an implant. Sandblasting and tricalcium phosphates spraying form a thin layer of bioceramics on the surface, giving durability and bioactivity to the implant [31]. In our work, more adherent cells were on control implants with anodization treatment.

This work studied how different methods of processing implants for cleaning from bacterial biofilm and protection from peri-implantitis affect the implant’s osseointegration. Although the development of treatment technologies using dental implants is advancing, many implant rejections are associated with insufficient osseointegration in the early stages after implantation and the development of peri-implantitis several years after implantation [32]. The post-implantation fate of an implant depends mainly on the surface structure, which includes topography, roughness, and surface biochemistry [23]. To create a microrelief, implants are processed by various physical and chemical methods, such as blasting, plasma spraying, acid etching, hydrogen peroxide treatment, etc.

The cause of peri-implatitis in patients with implants is still controversial. In particular, data on the effect of implant surface roughness on the incidence of peri-implant mucositis and peri-implantitis in humans is still limited [33]. Although titanium particles from implant surfaces are found in hard and soft peri-implant tissues, their role in the pathogenesis of peri-implantitis is also unclear [34]. Studies suggest that peri-implant mucositis is not associated with implants or surface roughness. [33,35]. The results of a clinical study including three systems of implants showed no difference in the incidence of peri-implantitis depending on the surface and design of the implant for 13 years [36].

According to some reports, the main reason for developing mucositis and peri-implantitis is the accumulation of pathogenic microflora, mainly spirochetes and Gram-negative anaerobes. With a decrease in immunity, the number of pathogenic microorganisms prevails, which provokes the development of various inflammatory processes. The inflammatory process is superficial in the case of mucositis and is deeper with peri-implantitis.

Currently, there are no actual data about the optimal parameters of the surface roughness of implants. Average isotropic roughness is considered the most suitable for osseointegration. Too high a roughness of the implant surface facilitates the adhesion of cells on the surface and bacteria, which is one of the reasons for peri-implantitis [27].

Four methods are used to clean the isolated from patients’ implants: diamond dental bur, titanium brush, laser, and air-flow.

In work [37], it was shown that the lowest risk of peri-implantitis was when using a dental bur for cleaning. However, this cleaning method significantly reduces osseointegration, as shown in our work: on implants treated with a bur, the number of adhered mesenchymal stem cells was minimal. Almost the same small number of attached cells was on the implant surface treated with a titanium brush. According to the SEM results, these two cleaning methods completely change the topography and surface roughness, removing nanostructures.

Air-flow treatment reduced surface roughness, decreasing the number of adherent cells, which was especially noticeable on XIVE Sirona Dentsply with an SLA surface (sandblasting combined with acid etching).

The maximum number of adherent cells close to control values was after laser treatment. At the same time, laser cleaning implants are very effective as antibacterial and safe for the tissues around the implant [38]. SEM has shown that this method of cleaning implants is the gentlest concerning the surface structure.

At the same time, other approaches to prevent peri-implantitis should be taken into account, for which, in particular, the use of proactive strategies has begun to maintain a balanced oral microbiota as much as possible [39], thus reducing the development of the inflammatory process. This approach suggests an increase in the effectiveness of the treatment of peri-implantitis without negatively impacting healthy tissues [40]. When using minimally invasive instrumentation, the frequency of soft tissue reorganization is reduced, and the balance of the bacterial biota present in the gingival sulcus and the gingival sulcus fluid is restored. Such minimally invasive effective methods of peri-implantitis therapy include ozone therapy [41], air polishing with glycine powder [42], and erythritol powder [43].

## 5. Conclusions

The implant surface cleaning method has a sufficient impact on peri-implantitis treatment. Our experiments demonstrated that laser treatment did not destroy the implant’s surface, so it maintains the capability to absorb osteogenic cells, such as MSC, for further division and differentiation. In general, to reduce the development of peri-implantitis, a combination of methods is required, both cleaning implants from a biofilm with pathogenic microbiota and restoring normal microbiota using probiotics and paraprobiotics [44].

## Figures and Tables

**Figure 1 dentistry-11-00030-f001:**
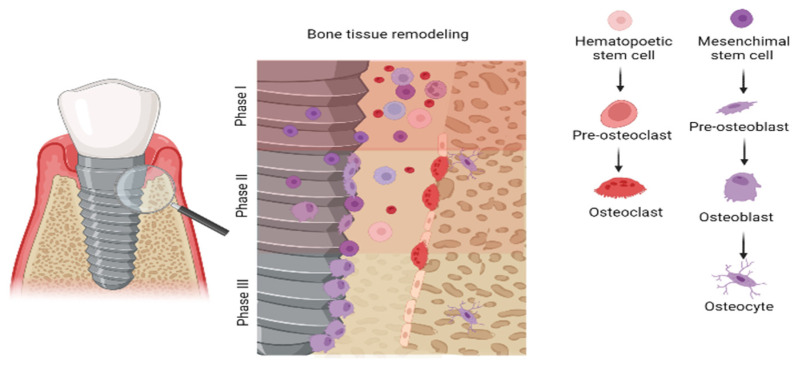
Mechanism of bone tissue remodeling after peri-implantitis treatment. The figure was created with BioRender.com.

**Figure 2 dentistry-11-00030-f002:**
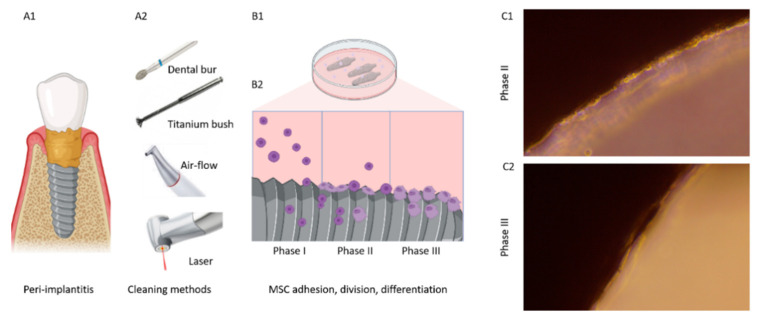
Principal scheme of the experiment: implants isolated from peri-implantitis (**A1**) were treated with different cleaning methods (**A2**) and cultured in an MSC-containing hydrogel-based culture medium (**B1**). Mesenchymal stem cells attached to the surface (Phase I, (**B2**)), divided (Phase II, (**B2**,**C1**)), and differentiated (Phase III, (**B2**,**C2**)).

**Figure 3 dentistry-11-00030-f003:**
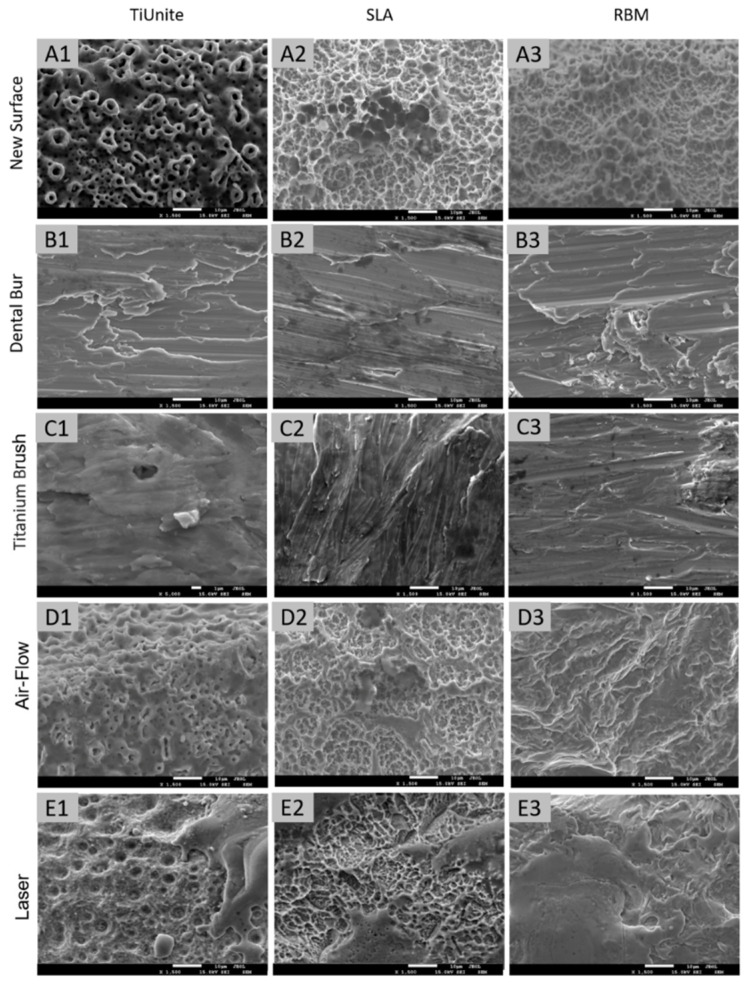
Impact of the cleaning method on the implant surface. Scanning electron microscopy (SEM) imaging of the surfaces of new untreated (**A1**–**A3**) implants TiUnite (**A1**), SLA (**A2**), and RBM (**A3**), as well as the implants from the same, manufacturers isolated from the patients with peri-implantitis and cleaned with different methods: dental bur (**B1**–**B3**), titanium brush (**C1**–**C3**), air-flow (**D1**–**D3**), and laser (**E1**–**E3**).

**Figure 4 dentistry-11-00030-f004:**
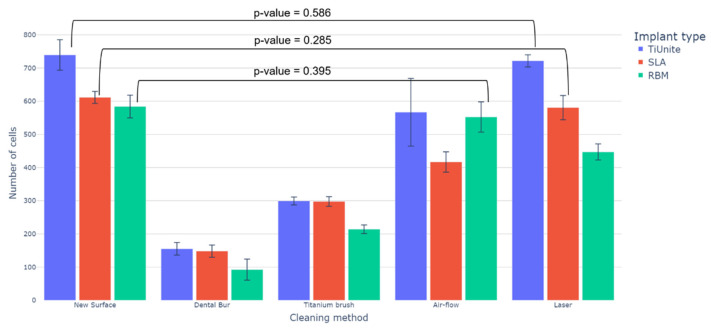
Mesenchymal stem cells adhesion on the surfaces of the differently treated implants. The most insignificant differences according to Welch’s t-test pairs of groups for (1) “TiUnite” are ‘New Surface’ and ‘Laser’ (*p*-value = 0.586); for (2) “SLA”—‘New Surface’ and ‘Laser’ (*p*-value) = 0.285; for (3) “RBM”—‘New Surface’ and ‘Air-flow’ (*p*-value = 0.395).

**Figure 5 dentistry-11-00030-f005:**
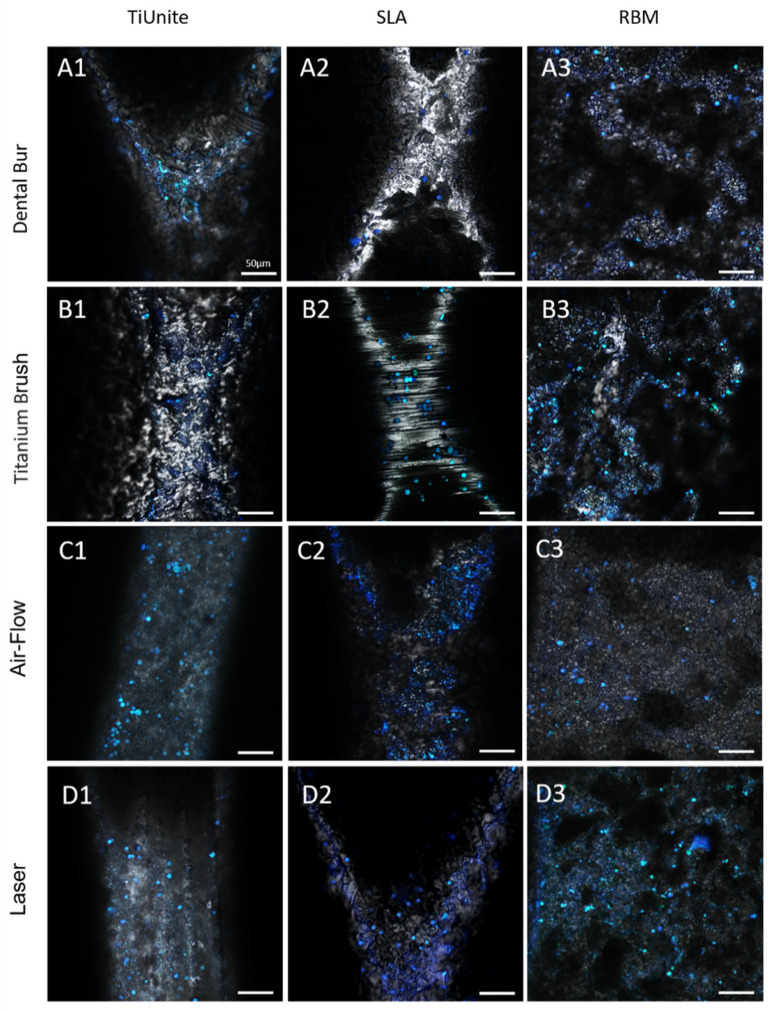
Mesenchymal stem cells growing on the surfaces of the TiUnite (**A1**), SLA (**A2**), and RBM (**A3**) implants isolated from the patients with peri-implantitis and treated with dental bur (**A1**–**A3**), titanium brush (**B1**–**B3**), air-flow (**C1**–**C3**), and laser (**D1**–**D3**).

## Data Availability

The data presented in this study are available on request from the corresponding author.

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
