# Peer review of "Laser Cleaning Improves Stem Cell Adhesion on the Dental Implant Surface during Peri-Implantitis Treatment"

_dentistry, 2023, doi:10.3390/dj11020030_

Round 1

Reviewer 1 Report

MANUSCRIPT OF CONSIDERABLE INTEREST FOR THE DENTAL SECTOR, IN PARTICULAR FOR ORAL SURGERY, BEFORE EVALUATING THE POSSIBILITY OF PUBLICATION, IT NEEDS A MAJOR REVISION.

ABSTRACT: HIGHLIGHTING OF STATISTICALLY SIGNIFICANT DATA IN ADDITION.

KEY WORDS, WELL DESCRIBED.

INTRODUCTION: WELL DESCRIBED, ONLY THE EVALUATION OF THE ORAL MICROBIOTA IN THE IMPLANT PATIENT ALREADY STUDIED IN THE RESEARCH GROUP OF PROF SCRIBANTE IS MISSING

THE VERY CONFUSING RESULTS HIGHLIGHT STATICLY SIGNIFICANT DATA SO THE READER CAN DETERMINE THE DIFFERENCES AT A FIRST SIGHT.

MATERIALS AND METHODS: HOW WAS THE SAMPLE SIZE CALCULATED?

DISCUSSION: ADD THE MINIMALLY INVASIVE APPROACH, THE USE OF ALL THE MINIMALLY INVASIVE SYSTEMS ALREADY STUDIED BY PROF SCRIBANTE'S RESEARCH GROUP AS FUTURE GOALS

10.3390/microorganisms10040675

CONCLUSION, ADD A PROACTIVE ACTION THROUGH NATURAL SUBSTANCES

Author Response

Dear authors,

Thank you for submitting your manuscript titled "Laser cleaning improves stem cell adhesion on the dental implant surface during peri-implantitis treatment".

Even though the topic is interesting and peri-implantitis is a major concern in modern dentistry, the manuscript you have submitted must be revised in many ways.

Here are the crucial points and flaws of the presented work:

  1. Language use and style has to be proofread by a native speaker. This manuscript requires extensive editing.

Dear Editor!

Thank you very much for your time and critical review. Please find below our pion-by -point response.

Answer: English language has been corrected using editing service.

  1. The materials and methods: sample size. The number of implants used in the study? 

Answer: Thank you very much for the comment. Data on the number of implants used in the studies are inserted into Matetials and Methods section.

  1. The results section should be completely revised. Most of the text written in this section belongs to materials and methods or discussion. The results written this way are certainly not clearly presented. The statements such as: "The surfaces of dental implants play a crucial role in osseointegration..." should not be a part of this section.

Answer: Thank you very much for the comment. The section has been substantially modified. The relevant parts of the text have been moved to Methods and Discussions sections. Data on the number of implants used in the studies are inserted into the results. Differentiation results on MSC were included.

  1. The discussion is written in a good manner, with minor corrections needed in formulation of the subsections. 

Considering the aforementioned facts, this manuscript is not suitable for publication before major revisions. 

Kind regards. 

MANUSCRIPT OF CONSIDERABLE INTEREST FOR THE DENTAL SECTOR, IN PARTICULAR FOR ORAL SURGERY, BEFORE EVALUATING THE POSSIBILITY OF PUBLICATION, IT NEEDS A MAJOR REVISION.

ABSTRACT: HIGHLIGHTING OF STATISTICALLY SIGNIFICANT DATA IN ADDITION.

Answer: Thank you very much for the comment. Abstract has been modified; the required information added.

KEY WORDS, WELL DESCRIBED.

INTRODUCTION: WELL DESCRIBED, ONLY THE EVALUATION OF THE ORAL MICROBIOTA IN THE IMPLANT PATIENT ALREADY STUDIED IN THE RESEARCH GROUP OF PROF SCRIBANTE IS MISSING

Thank you very much for the comment! Descriptions of the oral microbiota in patients’ implants was added in accordance with the review by prof. Scribante.

THE VERY CONFUSING RESULTS HIGHLIGHT STATICLY SIGNIFICANT DATA SO THE READER CAN DETERMINE THE DIFFERENCES AT A FIRST SIGHT.

Thank you very much for the comment! Significance of differences is added in Fig. 4

MATERIALS AND METHODS: HOW WAS THE SAMPLE SIZE CALCULATED?

Thank you very much for the comment! Data on the number of implants used in the studies are inserted into the section

DISCUSSION: ADD THE MINIMALLY INVASIVE APPROACH, THE USE OF ALL THE MINIMALLY INVASIVE SYSTEMS ALREADY STUDIED BY PROF SCRIBANTE'S RESEARCH GROUP AS FUTURE GOALS

Thank you very much for the idea. Data on the use of minimally invasive systems studied by prof. Sckribante were added.

10.3390/microorganisms10040675

CONCLUSION, ADD A PROACTIVE ACTION THROUGH NATURAL SUBSTANCES

Thank you very much for the comment. Data on proactive action with natural substances were added.

Reviewer 2 Report

Dear authors,

Thank you for submitting your manuscript titled "Laser cleaning improves stem cell adhesion on the dental implant surface during peri-implantitis treatment".

Even though the topic is interesting and peri-implantitis is a major concern in modern dentistry, the manuscript you have submitted must be revised in many ways.

Here are the crucial points and flaws of the presented work:

1. Language use and style has to be proofread by a native speaker. This manuscript requires extensive editing.

2. The materials and methods: sample size. The number of implants used in the study? 

3. The results section should be completely revised. Most of the text written in this section belongs to materials and methods or discussion. The results written this way are certainly not clearly presented. The statements such as: "The surfaces of dental implants play a crucial role in osseointegration..." should not be a part of this section.

4. The discussion is written in a good manner, with minor corrections needed in formulation of the subsections. 

Considering the aforementioned facts, this manuscript is not suitable for publication before major revisions. 

Kind regards. 

Author Response

(The authors gave the same response as above.)

Round 2

Reviewer 1 Report

The Manuscript has been revised correctly, please replace references 44 and 45 with the following:

https://doi.org/10.3390/app12062800

Author Response

Dear Reviewer, thank you very much for your work and time. 

We have corrected the English language.

We replaced references 44 and 45 with the one you recommended. 

Sincerely, Anna Kichkailo 

Reviewer 2 Report

Dear authors,

I find this manuscript suitable for publication in presented form.

Kind regards

Author Response

Dear Reviewer,

Thank you very much!

The English language has been corrected.

Sincerely, 

Anna Kichkailo